# Unifying Evidence on Delay Discounting: Open Task, Analysis Tutorial, and Normative Data from an Italian Sample

**DOI:** 10.3390/ijerph19042049

**Published:** 2022-02-11

**Authors:** Sara Garofalo, Luigi A. E. Degni, Manuela Sellitto, Davide Braghittoni, Francesca Starita, Sara Giovagnoli, Giuseppe di Pellegrino, Mariagrazia Benassi

**Affiliations:** 1Department of Psychology, University of Bologna, 47521 Cesena, Italy; luigialbert.degni@studio.unibo.it (L.A.E.D.); davide.braghittoni2@unibo.it (D.B.); francesca.starita2@unibo.it (F.S.); sara.giovagnoli@unibo.it (S.G.); g.dipellegrino@unibo.it (G.d.P.); mariagrazia.benassi@unibo.it (M.B.); 2Department of Comparative Psychology, Institute of Experimental Psychology, Heinrich-Heine University Düsseldorf, 40225 Düsseldorf, Germany; manuela.sellitto@hhu.de

**Keywords:** delay discounting, normative data, ROC curves, criterion validity, medial orbitofrontal cortex, impulsivity

## Abstract

Despite the widespread use of the delay discounting task in clinical and non-clinical contexts, several task versions are available in the literature, making it hard to compare results across studies. Moreover, normative data are not available to evaluate individual performances. The present study aims to propose a unified version of the delay discounting task based on monetary rewards and it provides normative values built on an Italian sample of 357 healthy participants. The most used parameters in the literature to assess the delay discount rate were compared to find the most valid index to discriminate between normative data and a clinical population who typically present impulsivity issues, i.e., patients with a lesion to the medial orbitofrontal cortex (mOFC). In line with our hypothesis, mOFC patients showed higher delay discounting scores than the normative sample and the normative group. Based on this evidence, we propose that the task and indexes here provided can be used to identify extremely high (above the 90th percentile for hyperbolic k or below the 10th percentile for AUC) or low (below the 10th percentile for hyperbolic k or above the 90th percentile for AUC) delay discounting performances. The complete dataset, the R code used to perform all analyses, a free and modifiable version of the delay discounting task, as well as the R code that can be used to extract all indexes from such tasks and compare subjective performances with the normative data here presented are available as online materials.

## 1. Introduction

Delay of gratification comprises a set of motivational and cognitive mechanisms that lead to larger later rewards as compared to immediate or closer rewards [1,2,3]. In this sense, it is intended as a form of regulatory process that involves the ability to control actions and feelings and the implementation of self-control strategies, which are necessary for the execution of difficult-to-achieve intentions [1,4]. The delay of gratification paradigm primarily measures the subjective preference for outcomes, and it has shown that most individuals ideally prefer larger later outcomes to smaller sooner outcomes when both instances are delayed in the future to some extent [5]. 

A related line of research in decision-making, however, has shown that when individuals make choices between actual options, they actually choose more often smaller rewards as the delay to their delivery becomes shorter—a phenomenon well-known as delay discounting or temporal discounting. In other words, delay discounting indicates a tendency to prefer smaller sooner rewards to larger delayed ones [6,7,8,9,10]. More precisely, delay discounting indicates the tendency to diminish the subjective value of a reward as a function of the time to its delivery: the further in the future, the lower its subjective value. To test this phenomenon, delay discounting tasks typically require individuals to choose between two options that vary in the amount (and often type) of reward and the temporal delay until delivery (e.g., intertemporal choice) [6,8,9,10]. For example, participants may have to choose between receiving EUR 50 now and EUR 100 in six months, or 1 candy bar now and 10 candy bars in two weeks. 

An increased preference for smaller but sooner (especially immediate) rewards is considered as an index of impulsivity and low self-control [9,11,12]. Accordingly, increased delay discounting has been proposed as a trans-disease process [13,14] and has been documented in several clinical populations typically associated with impulsive behavior [15], such as drug addiction [16,17,18,19,20], attention deficit hyperactivity disorder [21], compulsive gambling [22], and obesity [23,24,25,26]. However, delay discounting is also relevant in contexts that go beyond the pure clinical practice. Increased delay discounting has indeed been proposed as a useful marker of poor behavioral patterns in situations where, in fact, favoring distant goals over immediate gratification is extremely crucial, such as climate change, welfare policies action, and financial decisions [15,27,28,29,30,31].

Neuroimaging and lesion evidence converge to indicate that the medial orbitofrontal cortex (mOFC)—a region crucially involved in the attribution of “common currency” subjective value to rewards and in impulsive conduct [32,33,34]—is necessary for delayed gratification [35,36,37,38,39]. In line with this, patients with a lesion to the mOFC—as compared to control patients and healthy participants—tend to show an increased preference for smaller immediate rewards over larger delayed ones, regardless of the kind of reward at stake [8,40,41,42]. 

Delay discounting is relatively easy to measure, also in light of the large variety of tasks available in the literature. However, while this could be seen as an advantage, it also limits the direct comparison between studies. Different procedures have been used over the years to capture delay discounting in human behavior. From “fill-in-the-blank” or “matching” questions [43] to binary choices over a fixed-set of options [18] and binary choices with an underlying staircase design optimization procedure [40,44]. All such tasks have proven to reliably detect to a certain degree the typical (or atypical, when dysregulated) decay in subjective value for monetary and other rewards [20,38,45,46,47]. Nevertheless, very few studies compared different methodologies or their impact on the detection of inter-individual differences [45,48,49]. Crucially, normative data to evaluate individual performances are not currently available.

The present study aims to fill this gap by proposing a version of the delay discounting task based on monetary reward and providing percentile equivalents from an Italian sample of 357 healthy participants, stratified by sex, age, and education. To this aim, we chose to adopt a monetary version of the delay discounting task based on a staircase procedure that reliably detected delay discounting in healthy and pathological populations regardless of minor variations in the procedure or the model used to fit participants’ choices [25,40], as also confirmed by replication studies [42]. Crucially, we chose a task version that could be able to (a) provide enough data points (and, thus, indifference points) to fit the hyperbolic and exponential models and (b) that could be easily and meaningfully completed by participants in a few minutes, to overcome the crucial issue of fatigue in critical populations, such as patients or elderly participants.

The delay discount rate (i.e., the rate at which the subjective value of a reward decays with time) was assessed through the most used indexes in literature: the hyperbolic and exponential delay discounting parameters (k) [50,51], and the area under the empirical discounting curve (AUC) [52]. The parameters extracted from healthy participants were then contrasted and compared with those obtained from patients with brain lesions to the mOFC and a control group of patients. We expected mOFC patients to present higher delay discounting than healthy participants as well as control patients, in line with previous evidence [8,40,41,42]. Such differences will be used to suggest the strongest index (hyperbolic k, exponential k, or AUC) to indicate the presence of a particularly high delay discount rate (at least in this population). 

The complete dataset, the R code used to perform all analyses, a free and modifiable version of the delay discounting task, as well as the R code that can be used to extract all indexes from such tasks and compare subjective performances with the normative data here presented are available at https://osf.io/65bg7/ (accessed: 22 November 2021).

## 2. Materials and Methods

### 2.1. Participants

*Normative sample.* Healthy participants were randomly recruited by word of mouth and advertisement from the Italian population resident in Veneto, Marche, Lombardia, Campania, Toscana, and Emilia Romagna regions. Inclusion criteria involved having no history of neurological or psychiatric disorders, brain injury, epilepsy, drug abuse, or use of psychoactive drugs, and being Italian native speakers. Three hundred and fifty-seven volunteers met the eligibility criteria and were all included in the study (i.e., no participant meeting these criteria was excluded from the analysis). Participants were stratified by sex, age, and education (see Table 1 for demographics). 

*Patients with a lesion to the medial orbitofrontal cortex (mOFC).* Ten patients with brain damage predominantly involving the mOFC were included in the sample. Data from seven of these patients were obtained from a previous study using the same task [40]. Demographics are reported in Table 1. Patients were selected based on the location of their lesion as reported from magnetic resonance imaging (MRI) or computerized tomography (CT). These patients reported lesions encompassing the medial one-third of the orbital surface and the ventral one-third of the medial surface of the frontal lobe, as well as the white matter subjacent to these regions. Lesions were the results of the rupture of either aneurysm of the anterior communicating artery, traumatic brain injury, or meningioma. Lesions were bilateral in (8 cases, asymmetrically more prominent on the right in 3 cases and on the left in 2 cases) or lateralized on the right hemisphere (2 cases).

*Patients with a lesion to a non-frontal control site (nonFC).* Thirteen patients presented brain damage that did not involve the mesial orbital/ventromedial prefrontal cortex and frontal pole and spared the amygdala in both hemispheres were included in this sample. Lesions were unilateral in all cases and were all caused by ischemic/hemorrhagic stroke or meningioma. Demographics are reported in Table 1. Lesion sites included the lateral aspect of the temporal lobe and adjacent white matter (in 5 cases), the inferior parietal lobule (3 cases), the medial occipital area (1 case), the lateral occipitoparietal junction (2 cases), the cerebellum (1 case), or the temporal cortex (1 case). 

All patients (mOFC and nonFC) were in the stable phase of recovery, were not taking psychoactive drugs, and had no other diagnosis likely to affect cognition or interfere with participation in the study (e.g., significant psychiatric disease, alcohol abuse, history of cerebrovascular disease). All patients meeting these criteria were recruited at the Centre for Studies and Research in Cognitive Neuroscience of the University of Bologna (Cesena, Italy) and included in the statistical analysis.

All participants gave their formal consent to participate in the study, which was conducted following the ethical standards of the 1964 Declaration of Helsinki and approved by the ethical committee of the Department of Psychology of the University of Bologna. 

### 2.2. Delay Discounting Task

The task mirrors the one used by Sellitto and colleagues [25,40] using money as a reward (Figure 1). Participants chose between a hypothetical amount of money to be received immediately and a hypothetical amount of money to be received after some specified delay. There were six possible delays, each presented five times: 2 days, 2 weeks, 1 month, 3 months, 6 months, and 1 year. The order of blocks of choices for each delay was randomized. Within each block (five choices), the delayed amount was always EUR 40, while the amount of the immediate reward was adjusted based on the participant’s choices using a staircase procedure that converged on the amount of the immediate reward that was equal, in terms of subjective value, to the delayed reward (Du et al., 2002). The first choice was always between a delayed amount of EUR 40 and an immediate amount of EUR 20. If the participant chose the immediate reward, then the amount of the immediate reward was decreased on the next trial; if the subject chose the delayed reward, then the amount of the immediate reward was increased on the next trial. The size of the adjustment in the immediate reward decreased with successive choices: the first adjustment was half of the difference between the immediate and the delayed reward, whereas for subsequent choices it was half of the previous adjustment [53]. This procedure was repeated until the subject made five choices for each delay, after which a new series of five choices with a different delay began. Following this procedure, the immediate amount presented at each trial represents the best guess as to the subjective value of the delayed reward. Therefore, the immediate amount that would be presented on the sixth trial of a delay block is considered as the estimate of the subjective value of the delayed reward at that delay [25,40]. A free and modifiable version of the task is available at https://osf.io/65bg7/ (accessed: 22 November 2021).

### 2.3. Data Analysis

The complete dataset and the R code used to perform all analyses are available at https://osf.io/65bg7/ (accessed: 22 November 2021).

### 2.4. Estimation of the Delay Discount Rate

The three most used indices were taken from the literature to assess the delay discount rate (i.e., the rate at which the subjective value of a reward decays with delay): the hyperbolic and the exponential discounting parameter (*k*) (Mazur, 1987; Rachlin et al., 1991; Green and Myerson, 2004; Fellows and Farah, 2005), and the area under the empirical discounting curve (AUC) (Myerson et al., 2001).

*Estimation of k.* Hyperbolic [SV = 1/(1 + kD)] and exponential functions [SV = ekD] were fitted to the data, separately, to determine the k constant of the best fitting delay discounting function using a nonlinear least-squares algorithm (as implemented in Statistica, StatSoft). SV represents the subjective value (expressed as a fraction of the delayed amount) and D the delay (in days). The larger the value of k, the steeper the discounting function, the more participants were inclined to choose small immediate rewards over larger delayed rewards [40]. Following log-transformation, both hyperbolic and exponential k constants were normally distributed, and therefore comparisons were performed using parametric statistical tests when appropriate. R^2^ values were used to estimate the fit of both functions with individual data. The Wilcoxon rank-sum test (with continuity correction) reported a significantly higher fit (W = 72304, p = 0.002) for the hyperbolic function (R^2^ median = 0.75, IQR = 0.46, min = 0.00, max = 0.99) than for the exponential function (R^2^ median = 0.64, IQR = 0.66, min = 0.00, max = 0.99). Thus, the hyperbolic k was selected for all subsequent analyses. 

*Estimation of AUC.* Delays were expressed as a proportion of the maximum delay (360 days), and subjective values were expressed as a proportion of the delayed amount (40 units). Delays and subjective values were then plotted as x and y coordinates, respectively, to construct a discounting curve. Vertical lines were drawn from each x value to the curve, subdividing the area under the curve into a series of trapezoids. The area of each trapezoid was calculated as (x_2_ − x_1_)( y_1_ − y_2_)/2, where x_1_ and x_2_ are successive delays, and y_1_ and y_2_ are the subjective values associated with these delays [52]. The AUC is the sum of the areas of all the trapezoids and varies between 0 and 1. The smaller the AUC, the steeper the delay discounting, the more participants were inclined to choose small immediate rewards over larger delayed rewards [40]. AUC values were normally distributed; therefore, comparisons were performed using parametric statistical tests when appropriate.

### 2.5. Data Distribution

Data distribution was determined by estimating the Bayesian information criterion (BIC) and the Akaike information criterion (AIC) for the Gaussian and Cauchy distributions. In all cases, the Gaussian distribution showed the best fit (i.e., lowest value) (Table 2), so a general linear model was used for the subsequent analysis. 

### 2.6. Inconsistent Preferences

When estimating delay discounting it is important to rule out the possibility that group differences could be attributed to other factors, such as a different level of nonsystematic response patterns [54], i.e., inconsistent preferences. Hence, we counted the number of inconsistent preferences in each participant and compared them amongst groups. By definition, delay discounting should result in a monotonic decrease of the subjective value of the future outcome with delay [40,54]. That is, if *R*1 is the subjective value of a reward *R* delivered at delay *t*1, *R*2 is the subjective value of *R* delivered at delay *t*2, and *t*2 > *t*1, then it is expected that *R*2 < *R*1. As a consequence, subjects exhibit an inconsistent preference when the subjective value of the future outcome at a given delay is greater than that at the preceding delay, i.e., *R*2 > *R*1 [40,54]. To allow for some variability in the data, we considered as indicative of inconsistent preferences only those data points in which the subjective value of a reward overcame that at the preceding delay by a value of ≥20% of the future outcome (i.e., *R*2 ≥ *R*1 + *R*/5), in line with previous reports [40,54].

As we aimed at testing the absence of difference between groups, the individual mean number of inconsistent preferences was compared across groups using a Bayesian ANOVA, reported as the amount of evidence of the null hypothesis (absence of difference) over the alternative hypothesis (presence of difference), i.e., BF_01_.

### 2.7. Normative Data Estimation

To test whether sex, age, or education influence delay discounting, a general linear model was used. Two separate models were independently estimated using hyperbolic k or AUC as dependent variables and sex, age, and years of education and independent variables.

Percentile equivalents were subsequently stratified according to the resulting relevant variables and estimated with the R function “quantile” (type 8), which estimates sample quantiles corresponding to the given probabilities [55,56]. The smallest observation corresponds to a probability of 0 and the largest to a probability of 1. All sample quantiles are defined as weighted averages of consecutive order statistics. Sample quantiles of type i are defined by:Q[i](p) = (1 − γ) x[j] + γ x[j + 1]
where 1 ≤ i ≤ 9, (j − m)/n ≤ p < (j − m + 1)/n, x[j] is the jth order statistic, n is the sample size, the value of γ is a function of j = floor(np + m) and g = np + m − j, and m is a constant determined by the sample quantile type. Q[i](p) was considered a continuous function of p, with gamma = g and m given below. The sample quantiles were obtained by linear interpolation between the points (p[k],x[k]) where x[k] is the kth order statistic, m = (p + 1)/3. p[k] = (k − 1/3)/(n + 1/3), and p[k] = ~median[F(x[k])]. The resulting quantile estimates are approximately median-unbiased regardless of the data distribution (for further details see Hyndman and Fan, 1996).

### 2.8. Criterion Validity

Criterion validity concerns the ability of a test to differentiate between cases and controls. Separate generalized linear models were calculated on hyperbolic k and AUC, using in both cases the group as independent variable and sex and education as covariates (see Table 1 for the covariate choice).

To further confirm the efficacy of both hyperbolic k and AUC indexes to differentiate between normal and extreme performances in the delay discounting task, the scores from the normative sample were compared with the scores obtained from a clinical population known for presenting a high delay discount rate, i.e., patients with a lesion to the medial orbitofrontal cortex (mOFC), and a control group of patients with a lesion to a non-frontal site (nonFC), expected to report a performance coherent with the normative sample [8,40]. 

### 2.9. Receiver Operating Characteristic (ROC) Curves

To quantify the accuracy of the hyperbolic k and AUC as classifiers for mOFC patients, receiver operating characteristic (ROC) curves were used to estimate sensitivity and sensibility. ROC curves show the true-positive and false-positive rates associated with a range of possible cut-off points that distinguish patients from the normative sample. The true-positive rate (sensitivity) represents the proportion of mOFC patients that were correctly classified as dysfunctional. The false-positive rate (1-specificity) represents the proportion of mOFC patients that were incorrectly classified as functional. The optimal cut-off value was identified via the Youden index, corresponding to the score representing the maximum true-positive rate and minimum false-positive rate. 

### 2.10. Open Materials, Data, and Tutorials

The complete dataset, the R code used to perform all analyses, the delay discounting task, as well as the R code that can be used to extract all indexes from such tasks and compare subjective performances with the normative data here presented are available at https://osf.io/65bg7/ (accessed: 22 November 2021).

The original dataset (“DelayDiscountingNormativeData.xlsx”) contains all unprocessed data. The R code used to perform all analyses (“DelayDiscounting_norms.R”) on a step-by-step basis. The free and modifiable (e.g., in other languages) version of the delay discounting task (“DelayDiscountingTask.osexp”) is programmed in OpenSesame [57]. A further R code (“DelayDiscounting_tutorial.R”) contains a step-by-step tutorial that can be used to extract the relevant data from the output file generated by our version of the delay discounting task, calculate the necessary indexes, and compare a single performance with the normative data here presented. 

## 3. Results

### 3.1. Inconsistent Preferences

In line with our expectations, results indicated that the number of inconsistent preferences was small (HC m = 0.30, sd = 0.29; mOFC m = 0.33, sd = 0.33; nonFC m = 0.31, sd = 0.30) and comparable (BF_01_ = 7.04; error% = 0.03) across groups, replicating previous evidence [40].

### 3.2. Normative Data

Results from the general linear model revealed an influence of sex and education on both hyperbolic k and AUC scores (Table 3). Based on this evidence, percentile equivalents for hyperbolic k (Table 4) and AUC (Table 5) were stratified for sex and three education levels corresponding to primary (3–8 years), secondary (9–13 years), and bachelor’s or equivalent degree and above (14–23 years). These normative data aim for the identification of extreme performances. 

For hyperbolic k scores (Table 4), the bigger the value, the higher the delay discounting [40]. Thus, k scores falling below the 10th percentile indicate extremely low performance, while k scores falling above the 90th percentile indicate extremely high performance. For instance, any female with 9 to 13 years of education scoring lower than −7.57 should be considered as presenting an extremely low delay discounting; similarly, any female with 9 to 13 years of education scoring higher than 2.73 on the hyperbolic k should be considered as presenting an extremely high delay discounting (Table 4). 

The opposite interpretation should be applied to AUC. For AUC scores (Table 5), the smaller the value, the larger the delay discounting [40]. Thus, AUC scores falling below the 10th percentile indicate extremely high performance, while k scores falling above the 90th percentile indicate extremely low performance. For instance, any male with 9 to 13 years of education scoring lower than 0.08 should be considered as presenting an extremely high delay discounting; similarly, any male with 9 to 13 years of education scoring higher than 0.79 on the hyperbolic k should be considered as presenting an extremely low delay discounting (Table 5).

### 3.3. Criterion Validity

*Hyperbolic k*. Results showed higher scores for the mOFC group (mean = −2.33; sd = 1.59) than for the normative control group (mean = −3.79; sd = 2.27), although this difference was slightly above statistical significance (F_(1, 362)_ = 3.73; *p* = 0.054; part. η^2^ = 0.01). Both sex (F_(1, 362)_ = 9.74; *p* = 0.002; part. η^2^ = 0.03) and education (F_(2, 362)_ = 0.8.03; *p* = 0.0003; part. η^2^ = 0.04) reported a statistically significant influence on hyperbolic k. Bartlett’s homogeneity tests showed no evidence of differences amongst group variances (K^2^_(5)_ = 8.33; *p* = 0.13). These results show that the hyperbolic k can differentiate between performances in the normative and clinical samples. As evidenced in Figure 2, all but one male nonFC control patient with the highest level of education (level 3 = 14–23 years) showed comparable scores to those reported in the normative sample (i.e., within the 10th and 90th percentiles); mOFC were strongly differentiated from nonFC patients and the normative sample (above the 10th percentile) when considering a medium level of education (level 2 = 9–13 years) but less differentiated when considering the lowest (level 1 = 3–8 years) and highest levels of education (level 3 = 14–23 years). 

*AUC.* Results showed significantly lower scores (F_(1, 362)_ = 5.98; *p* = 0.015; part. η^2^ = 0.02) for the mOFC group (mean = 0.21; sd = 0.2) than for the normative control group (mean = 0.41; sd = 0.25). Both sex (F_(1, 362)_ = 7.59; *p* = 0.006; part. η^2^ = 0.02) and education (F_(2, 362)_ = 4.68; *p* = 0.01; part. η^2^ = 0.03) reported an influence on AUC. Bartlett’s homogeneity tests showed no evidence of differences amongst group variances (K^2^_(5)_ = 3.36; *p* = 0.37). These results indicate that the AUC can differentiate between performances in the normative and clinical samples.

As evidenced in Figure 3, all but two male nonFC control patients—one with education level 2 (9–13 years) and one with education level 3 (14–23 years)—showed comparable scores to those reported in the normative sample (i.e., within the 10th and 90th percentiles); mOFC were strongly differentiated from nonFC patients and the normative sample (above the 10th percentile) when considering a medium level of education (level 2 = 9–13 years) but less differentiated when considering the lowest (level 1 = 3–8 years) and highest levels of education (level 3 = 14–23 years). 

Overall, criterion validity results suggest that percentile equivalents based on the hyperbolic k and the AUC are similarly effective to differentiate between patients and controls. 

### 3.4. ROC Curves

Figure 4 shows the ROC curves for the hyperbolic k and AUC scores. The sensitivity and specificity curves were comparable for the hyperbolic k (area under the curve = 0.73; Youden index value = 0.51, cut-off = −2.28, false-positive rate = 0.19, true-positive rate = 0.7) and AUC (area under the curve = 0.73; Youden index value = 0.47, cut-off = 0.85, false-positive rate = 0.13, true-positive rate = 0.6). 

## 4. Discussion

The present study aimed to propose a unified version of the delay discounting task and provide normative data (percentile equivalents) from an Italian sample of 357 healthy participants—stratified by sex, age, and education—collected via a monetary-based delay discounting task. The delay discount rate (i.e., the rate at which the subjective value of a reward decays at increasing time delay) was assessed through the most commonly used parameters in literature (namely, hyperbolic k, exponential k, and AUC), which we compared to find the most valid index to discriminate between clinical and non-clinical (i.e., mOFC patients) populations. The k parameter [50,51] was calculated by fitting both hyperbolic and exponential functions. Results indicated a higher fit for the hyperbolic k, which was henceforth selected for further exploration. Percentile equivalents were calculated for both hyperbolic k and AUC and stratified according to the demographic variables showing to exert an influence, namely, sex and education for both hyperbolic k and AUC (Table 1). 

In this regard, despite age differences in delay discounting previously described [50,58,59], no influence of age either on the hyperbolic k or the AUC was found in the present study. However, it has to be noted that past evidence on this topic is quite mixed [50,58,59,60], with some authors concluding that higher delay discounting is associated with younger age and others reporting that higher delay discounting characterizes an older age. Findings from a larger and stratified sample, such as the one used in this study, seem to suggest that such contrasting results may be explained by individual variations present in the smaller and more diverse samples used in previous studies. The normative data were further compared with the scores from mOFC and a control group of patients to test for criterion validity. 

In line with our hypothesis and previous evidence [8,34,35,36,40], mOFC patients showed higher delay discounting scores than the normative sample and the control group. Hyperbolic k and AUC were similarly effective to identify extreme scores based on percentile equivalents (Figure 2 and Figure 3). ROC curves confirmed such observations (Figure 4), reporting higher sensitivity and specificity for the hyperbolic k, as compared to the AUC. Based on these analyses, we propose that the task and the normative data presented in this paper can be used to identify extremely high or low delay discounting performances. Since hyperbolic k higher scores indicate higher delay discounting [40], extremely low or high performances are indicated by k scores falling below the 10th percentile or above the 90th percentile, respectively. For instance, any male with 9 to 13 years of education scoring lower than −6.55 should be considered as presenting an extremely low delay discounting. In opposition, since AUC lower scores indicate higher delay discounting [40], extremely low or high performances are indicated by AUC scores falling above the 90th percentile or below the 10th percentile, respectively. For instance, any female with 9 to 13 years of education scoring lower than 0.02 should be considered as presenting an extremely high delay discounting (Table 5).

Some limitations characterize this study. First, both for the hyperbolic k and AUC, mOFC patients were best differentiated from control patients and the normative sample only when considering a medium level of education (level 2 = 9–13 years) but not when considering the lowest (level 1 = 3–8 years) and highest levels of education (level 3 = 14–23 years); thus, the normative data here proposed should be considered carefully if the patients or participants fall within very high or very low levels of education. Second, the patients’ sample is relatively small compared to the normative sample, although in line with the sample size usually achievable in a study for this kind of population. Notably, for both limitations, mOFC patients are only one of the possible populations known to report particularly extreme scores in temporal discounting [11,16,17,18,19,21,22,23,40,61,62,63,64,65,66,67,68,69]. Lastly, a limitation to the representativeness of the normative sample used in this study can be represented by the fact that the sample was mainly (although not exclusively) recruited in the northern regions of Italy (Veneto, Lombardia, and Emilia Romagna), although central (Marche, Toscana) and southern (Campania) regions were included (see Section 2.1). Crucially, the sample can be considered representative of the Italian adult population in terms of the three main characteristics considered, namely sex, age, and education age (see Table 2). Future studies may help clarify these issues by extending the evidence presented in this paper not only to other patients but also to normative samples from other nationalities. 

## 5. Conclusions

The present study proposes a unified version of the delay discounting task with monetary reward (see Section 2.10) and provides normative data based on two discount rate indexes, namely, the hyperbolic k and the AUC. In line with our hypothesis, mOFC patients showed steeper delay discounting than the normative sample and the control group of patients [8,34,35,36,40]. 

Based on this evidence, we submit that the task and the indexes here provided can be used to identify extremely high (above the 90th percentile for hyperbolic k or below the 10th percentile for AUC) or low (below the 10th percentile for hyperbolic k or above the 90th percentile for AUC) delay discounting performances.

In this sense, the present findings open a path to applications in both clinical and non-clinical contexts. Indeed, increased delay discounting is classically associated with maladaptive decision-making [3,4,5,6,7,8,9,10,11,12,13,14,15,16], whereas self-control is correlated with beneficial outcomes in terms of financial stability [70,71], academic achievement [72,73], social success [74,75], healthy living [76], and wellbeing [77,78]. In the same vein, research on delay of gratification revealed that high-delayers, as compared to low-delayers, show better inhibitory control (associated with more activated prefrontal cortex in tasks demanding increased inhibition) and that, during childhood, it can predict life outcomes [1]. The presence of a delay discount rate higher (or lower) than the normative population may, thus, hint to the need for further screening for impulsivity, compulsive behavior, and self-control issues [8,11,12,22,40,41,42,79]. 

## Figures and Tables

**Figure 1 ijerph-19-02049-f001:**
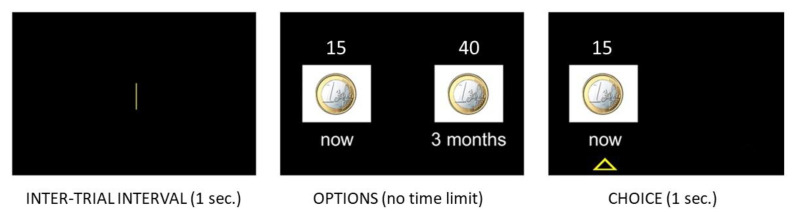
Graphical representation of the delay discounting task. In each trial, after a 1 s intertrial interval, participants chose between a small amount of money delivered immediately and a larger amount of money delivered after a delay. The preferred option remained highlighted for 1 s on the screen.

**Figure 2 ijerph-19-02049-f002:**
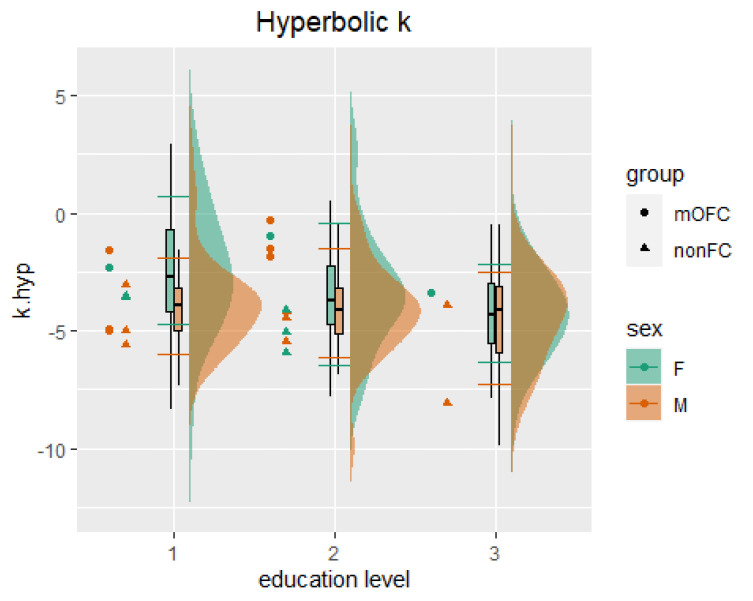
Comparison between normative and clinical data for the hyperbolic k. The figure shows the normative data distribution and boxplots, separated for the female (F) and male (M) subsamples and level of education (1 = 3–8 years; 2 = 9–13 years; 3 = 14–23 years). The horizontal lines on the boxplot represent the 10th (top) and 90th (bottom) percentiles.

**Figure 3 ijerph-19-02049-f003:**
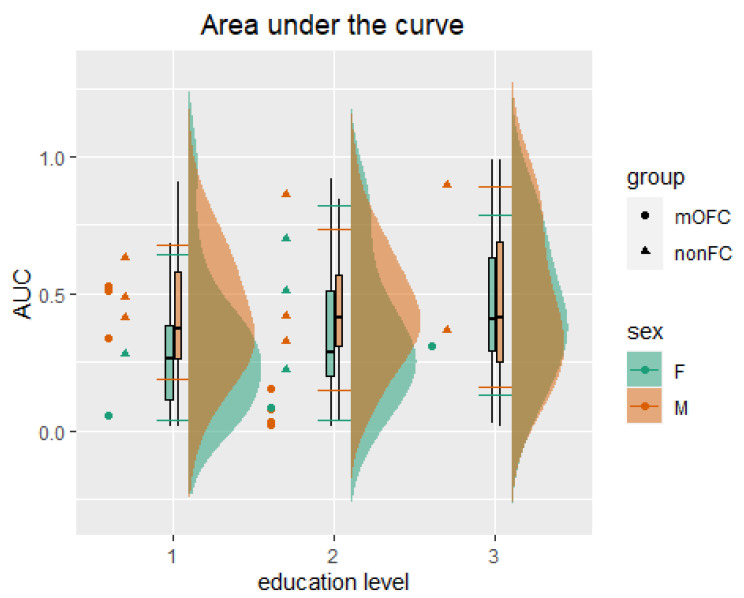
Comparison between normative and clinical data for the area under the curve (AUC). The figure shows the normative data distribution and boxplots, separated for the female (F) and male (M) subsamples and level of education (1 = 3–8 years; 2 = 9–13 years; 3 = 14–23 years). The horizontal lines on the boxplot represent the 10th (top) and 90th (bottom) percentiles.

**Figure 4 ijerph-19-02049-f004:**
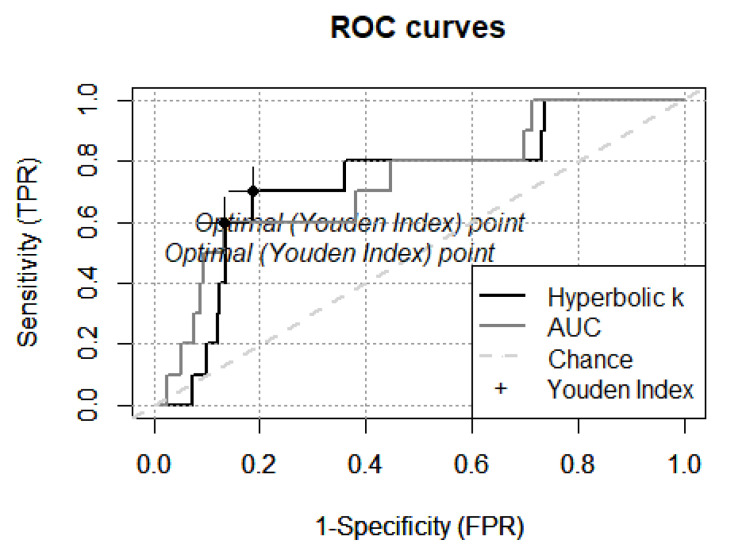
ROC curves for hyperbolic k and AUC. Sensitivity (true-positive rate, TPR) and 1-specificity (false-positive rate, FPR) curves for the hyperbolic k (in black) and AUC (in grey).

**Table 1 ijerph-19-02049-t001:** Demographic characteristics of the normative sample.

		Normative Sample (N = 357)	mOFC (N = 10)	nonFC (N = 13)
**Sex**	N (m/f)	187/170	7/3	5/8
**Age**	N (21–39/40–29/60–92 yrs)	91/140/126	0/3/7	1/5/7
	mean (sd)	52.18 (16.71)	59.8 (9.15)	58.8 (12.77)
	median (min–max)	54 (21–92)	62 (41–70)	60 (31–76)
**Education**	N (3–8/9–13/14–23 yrs)	102/128/127	4/5/1	5/6/2
	mean (sd)	12.69 (4.36)	10.8 (5.65)	11.38 (3.99)
	median (min–max)	13 (3–23)	12 (4–23)	13 (5–19)

**Table 2 ijerph-19-02049-t002:** Fitting values for the normal and Cauchy distributions to k and AUC scores.

	Gaussian		Cauchy	
	BIC	AIC	BIC	AIC
k	1610.75	1602.991	1646.329	1638.574
AUC	32.76 *	25.01 *	167.86	160.11

BIC = Bayesian information criterion. AIC = Akaike information criterion. * Best fit corresponding to the lowest BIC and AIC value.

**Table 3 ijerph-19-02049-t003:** Influence of sex, age, and education on hyperbolic k and AUC.

	Hyperbolic k			AUC		
	F (df)	*p*	Part. H^2^	F (df)	*p*	Part. H^2^
Sex	8.86 (1)	0.003 **	0.02	6.57 (1)	0.010 *	0.02
Age	0.05 (1)	0.819	1.48e-04	0.13 (1)	0.716	3.76e-04
Education	11.64 (1)	0.0007 ***	0.03	7.65 (1)	0.006 **	0.02

* *p* < 0.05; ** *p* < 0.01; *** *p* < 0.001; AUC = area under the curve.

**Table 4 ijerph-19-02049-t004:** Percentile equivalents for hyperbolic k scores.

Hyperbolic k						
Sex	F			M		
Education (Years)	3–8	9–13	14–23	3–8	9–13	14–23
**Percentile**						
**5**	−6.35	−7.57	−7.51	−6.51	−6.55	−7.59
**10**	−5.06	−6.62	−6.42	−6.03	−6.16	−7.32
**25**	−4.21	−4.80	−5.55	−5.08	−5.12	−6.00
**50**	−2.70	−3.73	−4.35	−3.94	−4.10	−4.13
**75**	−0.67	−2.19	−2.94	−3.20	−3.17	−3.10
**85**	0.35	−1.03	−2.38	−2.73	−2.20	−2.84
**90**	0.97	−0.18	−2.08	−1.75	−1.42	−2.37
**95**	2.38	2.73	−1.00	0.69	−0.42	−1.91

Percentile equivalents are stratified for sex and three education levels corresponding to primary (3–8 years), secondary (9–13 years), and bachelor’s or equivalent and above (14–23 years).

**Table 5 ijerph-19-02049-t005:** Percentile equivalents for AUC scores.

AUC						
Sex	F			M		
Education (Years)	3–8	9–13	14–23	3–8	9–13	14–23
**Percentile**						
**5**	0.02	0.02	0.04	0.02	0.08	0.08
**10**	0.04	0.03	0.12	0.18	0.14	0.16
**25**	0.11	0.20	0.29	0.26	0.30	0.25
**50**	0.26	0.29	0.40	0.37	0.41	0.41
**75**	0.38	0.52	0.66	0.59	0.57	0.70
**85**	0.46	0.72	0.77	0.66	0.69	0.85
**90**	0.66	0.82	0.79	0.71	0.74	0.89
**95**	0.97	0.87	0.89	0.79	0.79	0.91
**5**	0.02	0.02	0.04	0.02	0.08	0.08

Percentile equivalents are stratified for sex and three education levels corresponding to primary (3–8 years), secondary (9–13 years), and bachelor’s or equivalent and above (14–23 years).

## Data Availability

The complete dataset, the R code used to perform all analyses, a free and modifiable version of the delay discounting task, and the R code that can be used to extract all indexes from such tasks and compare subjective performances with the normative data here presented are available at https://osf.io/65bg7/ (accessed: 22 November 2021).

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
