# Peer review of "Unifying Evidence on Delay Discounting: Open Task, Analysis Tutorial, and Normative Data from an Italian Sample"

_ijerph, 2022, doi:10.3390/ijerph19042049_

Round 1

Reviewer 1 Report

The purpose of this study was to propose a ‘unified version’ of the delay discounting (DD) task by publishing a modifiable version of the task and providing normative data on DD task performance in a small sample of Italian adults. The focus and purpose of the study is novel and important and would be of interest to IJERPH readers. More importantly, the study represents an important advance in the use of DD as an assessment tool to be used in clinical settings. To that extent, the novelty and importance of the study are excellent and I appreciate the authors’ focus on this topic. There are many strengths to this study and work on this front is incredibly important and should be pursued. That said, I do have several concerns about the manuscript, which I outline here.

The literature review could benefit from drawing the reader’s attention more effectively to the broad relevance of DD to understanding human health problem behaviors. For example, I recommend adding some language that clarifies DD’s role as a trans-disease process (Amlung, et al., 2019; Bickel, et al., 2012, 2019). Although a bit dated, Mackillop’s (2011) meta-analysis about DD and addiction also is relevant. On line 38, the authors suggest that DD ‘characterizes’ several clinical populations, but I’m not sure that is the right language. It certainly is an important underlying factor, but does not ‘coherently’ characterize these complex phenomena. One important aspect of the paper is the promotion of a consistent procedure to measure DD but it would be helpful to characterize the various ways that DD is assessed in the literature. The authors might also consider discussing the limited research to date comparing different methods (e.g., Robles, et al. 2009), which would help buttress their arguments for the study.

Normative data for DD have too long been missing from the literature and I applaud the researchers’ efforts on this front. However, the procedures used to generate the study’s normative sample are potentially problematic since it is basically a convenience sample rather than one drawn from a randomly selected group of community members. However, those concerns might be offset by the novelty of the study overall. The use of both mOFC and nonFC groups is notable, but I am concerned about comparing groups with such differing sample sizes from the normative sample, especially given the potential confounds of sex and education identified by the authors. I’m not certain, but a better approach might be to match participants from the normative sample and then make comparisons. The authors describe testing for ‘discriminant validity’ but I don’t think this is correct (see Campbell & Fiske, 1959). They might actually be testing for concurrent validity, but this should be clarified.

The authors do a nice job of using an established discounting method for their task. One critically novel aspect of the study is the publication of the programming for the DD task they used. This is a wonderful advancement toward standardizing how DD tasks are presented to participants and I appreciate the authors’ effort! They might consider cutting out the exponential decay data, since hyperbolic decay is pretty standard; I’ll note, though, that hyperbola-like models (e.g., Green, Fry, & Myerson, 1994) are frequently used also. One issue that I’m concerned about is that the authors seem to be using raw k values for their parametric analyses. k values in DD are almost universally positively skewed and require log transformation to be normalized. AUC values tend to be more normally distributed and this seems to be reflected in Table 2 but the authors should ensure that the assumptions of their analyses are in fact met. What is also missing is any discussion of nonsystematic response patterns (Johnson & Bickel, 2008) that sometimes occur when using titration tasks like the one used here. Smith, et al. (2018) outline recommendations for reporting these data, which are especially relevant in the context of interpreting individual patterns of DD, which the authors aspire to do here.

The effort to provide normative data is notable and novel and should be pursued, especially in the context of identifying clinical problems (as the authors seem to try to do here), but the present study doesn’t seem to lead to relevant information that would clarify whether a particular pattern of discounting might be relevant to a clinical problem, such as a neurological lesion. It isn’t clear how simply pointing out that someone has a statistically high k or AUC score is helpful in terms of identifying problem outcomes. It could be that I am missing something and would gratefully accept evidence-based feedback on this front.

Although I have several concerns about how the data are presented and interpreted here, I think the work here is important and worth pursuing. If this particular paper is not accepted, I encourage the authors to continue their work down this very important path!

Author Response

Reviewer 1

  1. The literature review could benefit from drawing the reader’s attention more effectively to the broad relevance of DD to understanding human health problem behaviors. For example, I recommend adding some language that clarifies DD’s role as a trans-disease process (Amlung, et al., 2019; Bickel, et al., 2012, 2019). Although a bit dated, Mackillop’s (2011) meta-analysis about DD and addiction also is relevant. On From line 38, the authors suggest that DD ‘characterizes’ several clinical populations, but I’m not sure that is the right language. It certainly is an important underlying factor, but does not ‘coherently’ characterize these complex phenomena.

We thank the reviewer for this comment, as it gave us the opportunity to highlight the ubiquitous implications of the delay discounting. This included not only claryfing its role in clinical contexts, but also its impact outside clinical scenarios. We also agree with the reviewer that the term ‘characterizes’ was imprecisely used. We expanded references (in red in the revised manuscript) including those suggested (and some more) and rephrased as follows:

From line 38

Accordingly, increased delay discounting has been proposed as a trans-disease process [8,9] and has been documented in several clinical populations typically associated with impulsive behavior [10], like drug addiction [11–15], attention deficit hyperactivity disor-der [16], compulsive gambling [17], and obesity [18–21].

From line 43

Increased delay discounting has indeed been proposed as a useful marker of poor behav-ioral patterns in situations where, in fact, favoring distant goals over immediate gratifica-tion is extremely crucial, like climate change, welfare policies action, and financial deci-sions [10,22–26].

From line 416

The presence of unusually increased or reduced delay discount rates can be used, in both clinical and non-clinical contexts, as an indicator of impulsive behavior or low self-control that may require further attention. Indeed, high delay discounting is classical-ly associated with maladaptive decision-making [3–16], while self-control is correlated with beneficial outcomes in terms of financial stability [56,57], academic achievement [58,59], social success [60,61], healthy living [62], and wellbeing [63,64].

  1. One important aspect of the paper is the promotion of a consistent procedure to measure DD but it would be helpful to characterize the various ways that DD is assessed in the literature. The authors might also consider discussing the limited research to date comparing different methods (e.g., Robles, et al. 2009), which would help buttress their arguments for the study.

We thank the reviewer for this comment, which helped us in clarifying our aim as well as some methodological choices. The following text was added to the introduction:

From line 53

Delay discounting is relatively easy to measure, also in light of the large variety of tasks available in the literature. However, while this could be seen as an advantage, it also limits the direct comparison between studies. Different procedures have been used over the years to capture delay discounting in human behavior. From “fill-in-the-blank” or “matching” questions [38] to binary choices over a fixed-set of options [13] and binary choices with an underlying staircase design optimization procedure [35,39]. All such tasks have proven to reliably detect to a certain degree the typical (or atypical, when dysregulated) decay in subjective value for monetary and other rewards [15,33,40–42]. Nevertheless, very few studies compared different methodologies or their impact on the detection of inter-individual differences [40,43,44]. Crucially, normative data to evaluate individual performances are not currently available.

The present study aims to fill this gap by proposing a version of the delay discount-ing task based on monetary reward and providing percentile equivalents from an Italian sample of 357 healthy participants, stratified by sex, age, and education. To this aim, we chose to adopt a monetary version of the delay discounting task based on a staircase pro-cedure that reliably detected delay discounting in healthy and pathological populations regardless of minor variations on the procedure or the model used to fit participants’ choices [20,35], as also confirmed by replication studies [37]. Crucially, we chose a task version that could be able to (a) provide enough data points (and, thus, indifference points) to fit the hyperbolic and exponential models and (b) that could be easily and meaningfully completed by participants in a few minutes, to overcome the crucial issue of fatigue in critical populations, such as patients or elderly participants.

  1. The use of both mOFC and nonFC groups is notable, but I am concerned about comparing groups with such differing sample sizes from the normative sample, especially given the potential confounds of sex and education identified by the authors. I’m not certain, but a better approach might be to match participants from the normative sample and then make comparisons.

Since we agree with the reviewer that comparing groups with different sample sizes may pose some problems, we adopted, accordingly, a statistical model that can take into account, and correct for, possible imbalances amongst the variables inserted in the model, namely, sex, age, and education (see Huitema, B. 1980. The analysis of covariance and alternatives. New York: Wiley).

Moreover, when using Anova, the robustness of results despite possible imbalances is ensured as long as variances are homogeneous, which was the case for our analyses (see Wilcox, R. R., Char1in, V. L., & Thompson, K. L. 1986. New monte carlo results on the robustness of the anova f, w and f statistics. Communications in Statistics-Simulation and Computation, 15-4, 933-943).

Nevertheless, we acknowledge that we failed to specify this information in the results. We, thus, added the following text in the revised manuscript:

From line 317

Bartlett’s homogeneity tests showed no evidence of differences amongst group variances (K2(5) = 8.33; p = 0.13). 

From line 337

Bartlett’s homogeneity tests showed no evidence of differences amongst group variances (K2(5) = 3.36; p = 0.37).

  1. The authors describe testing for ‘discriminant validity’ but I don’t think this is correct (see Campbell & Fiske, 1959). They might actually be testing for concurrent validity, but this should be clarified.

We thank the reviewer for spotting this imprecision. We agree that our use of the term “discriminant validity” was imprecise and we appreciate the suggestion about the possibility of testing for concurrent validity instead. We decided to refer more generally to the term “criterion validity” (cfr here), which comprises both concurrent and predictive validity, as in our case no clear temporal criterion could be established. We rephrased the revised manuscript accordingly (see from line 242 and line 321).

  1. They might consider cutting out the exponential decay data, since hyperbolic decay is pretty standard; I’ll note, though, that hyperbola-like models (e.g., Green, Fry, & Myerson, 1994) are frequently used also.

We agree that it is no longer the standard measure, but exponential discounting is still present in the literature and widely used by some authors, thus we decided to keep it in the manuscript as (hopefully informative) additional evidence for comparison.

  1. One issue that I’m concerned about is that the authors seem to be using raw k values for their parametric analyses. k values in DD are almost universally positively skewed and require log transformation to be normalized. AUC values tend to be more normally distributed and this seems to be reflected in Table 2 but the authors should ensure that the assumptions of their analyses are in fact met.

We thank the reviewer for spotting this lack of information in our paper. K values were actually already log-transformed, as visible in the R file available on OSF as part of the paper (DelayDiscounting_norms.R, code lines 34-36). However, we did not mention such transformation when describing the estimation of k in the manuscript. We added the following text to the revised manuscript:

From line 175

Following log-transformation, both hyperbolic and exponential k constants were normally distributed, therefore comparisons were performed using parametric statistical tests when appropriate.

From line 192

AUC values were normally distributed, therefore comparisons were performed using parametric statistical tests when appropriate.

  1. What is also missing is any discussion of nonsystematic response patterns (Johnson & Bickel, 2008) that sometimes occur when using titration tasks like the one used here. Smith, et al. (2018) outFrom line recommendations for reporting these data, which are especially relevant in the context of interpreting individual patterns of DD, which the authors aspire to do here.

We happily welcomed the suggestion to add an analysis of non-systematic response patterns, which indeed can provide evidence to the robustness of our findings. In the revised manuscript, we added the following dedicated paragraphs to the Materials and methods and Results sections, respectively:

From line 204

Inconsistent preferences

When estimating delay discounting it is important to rule out the possibility that group differences could be attributed to other factors, like a different level of nonsystematic response patterns [49], i.e., inconsistent preferences. Hence, we counted the number of in-consistent preferences in each participant and compared them amongst groups. By defini-tion, delay discounting should result in a monotonic decrease of the subjective value of the future outcome with delay [35,49]. That is, if R1 is the subjective value of a reward R deliv-ered at delay t1, R2 is the subjective value of R delivered at delay t2, and t2>t1, then it is expected that R2<R1. As a consequence, subjects exhibit an inconsistent preference when the subjective value of the future outcome at a given delay is greater than that at the pre-ceding delay, i.e., R2 > R1 [35,49]. To allow for some variability in the data, we considered as indicative of inconsistent preferences only those data points in which the subjective value of a reward overcame that at the preceding delay by a value of ≥ 20% of the future outcome (i.e., R2 ≥ R1 + R/5), in line with previous reports [35,49]. As we aimed at testing the absence of difference between groups, the individual mean number of inconsistent preferences was compared across groups using a Bayesian Anova, reported as the amount of evidence of the null hypothesis (absence of difference) over the alternative hypothesis (presence of difference), i.e. BF01.

From line 279

Inconsistent preferences

In line with our expectations, results indicated that the number of inconsistent pref-erences was small (HC m = 0.30, sd = 0.29; mOFC m = 0.33, sd = 0.33; non-FC m = 0.31, sd = 0.30) and comparable (BF01=7.04; error% = 0.03) across groups, replicating previous evi-dence [35].

  1. The effort to provide normative data is notable and novel and should be pursued, especially in the context of identifying clinical problems (as the authors seem to try to do here), but the present study doesn’t seem to lead to relevant information that would clarify whether a particular pattern of discounting might be relevant to a clinical problem, such as a neurological lesion. It isn’t clear how simply pointing out that someone has a statistically high k or AUC score is helpful in terms of identifying problem outcomes. It could be that I am missing something and would gratefully accept evidence-based feedback on this front.

We agree that our argumentations could benefit from clarifications about the usefulness of detecting a particularly high or low discounting. We belied we improved this lack of information by expanding the introduction as indicated in the first response to this revision (see paragraphs from lines 37, 49, 410).

Moreover, we added the following text to the Discussion and Results sections to guide the understanding and interpretation of the results presented in our study (also in response to a similar point raised by reviewer 2 – question n. 15):

From line 290

These normative data aim to the identification of extreme performances.

For hyperbolic k scores (Table 4), the bigger the value, the higher the delay discount-ing [35]. Thus, k scores falling below the 10th percentile indicate extremely low perfor-mance, while k scores falling above the 90th percentile indicate extremely high perfor-mance. For instance, any female with 9 to 13 years of education scoring lower than -7.57 should be considered as presenting an extremely low delay discounting; similarly, any female with 9 to 13 years of education scoring higher than 2.73 on the hyperbolic k should be considered as presenting an extremely high delay discounting (Table 4).

The opposite interpretation should be applied to AUC. For AUC scores (Table 5), the smaller the value, the larger the delay discounting [35]. Thus, AUC scores falling below the 10th percentile indicate extremely high performance, while k scores falling above the 90th percentile indicate extremely low performance. For instance, any male with 9 to 13 years of education scoring lower than 0.08 should be considered as presenting an ex-tremely high delay discounting; similarly, any male with 9 to 13 years of education scor-ing higher than 0.79 on the hyperbolic k should be considered as presenting an extremely low delay discounting (Table 5).

From line 405

Based on these analyses, we propose that the task and the normative data presented in this paper can be used to identify extremely high or low delay discounting performances. Since for hyperbolic k higher scores indicate higher delay discounting [35], extremely low or high performances are indicated by k scores falling below the 10th percentile or above the 90th percentile, respectively. For instance, any male with 9 to 13 years of education scoring lower than -6.55 should be considered as presenting an extremely low delay dis-counting. In opposition, since for AUC lower scores indicate higher delay discounting [35], extremely low or high performances are indicated by k scores falling above the 90th percentile or below the 10th percentile, respectively. For instance, any female with 9 to 13 years of education scoring lower than 0.02 should be considered as presenting an ex-tremely high delay discounting (Table 5).

From line 411

The presence of a delay discount rate higher (or lower) than a normative population may inform clinical practice by suggesting the need for further screening for impulsivity, compulsive behavior, and self-control issues [6,7,17,70].

Reviewer 2 Report

This is an rather interesting and important study that aimed to establish the delay discounting task in Italian subjects. The manuscript, however, should be revised in terms of several important issues.

1), from the introduction, it is unclear how many different versions of the delay discounting task exist and why did the authors choose the current version for their study.

2), it is unclear how the subjects were recruited, how many were dropped, and how the authors estimated the sample size for the current study. More importantly, do the authors consider the current sample being representative of the Italian population?

3), an important limitation of the study is the very small sample size of the mOFC and nonFC groups. Given this sample size, the results of the classification are likely to be unreliable.

4), for the delay discounting task, the authors should provide a graphic illustration. Furthermore, why did the authors not use 5 or 10 years in their task, these two delays have been frequently used in previous studies.

5), not only significance, effect size measures should be reported, for instance, Table 3, section Discriminant validity on page 7, etc.

6), can the authors provide some explanation or interpretation of the data reported in Table 4 and 5 in the main text?

Author Response

Reviewer 2

  1. from the introduction, it is unclear how many different versions of the delay discounting task exist and why did the authors choose the current version for their study.

We thank the reviewer for this comment, which helped us in clarifying our aim as well as some methodological choices. The following text was added to the introduction:

From line 54

Delay discounting is relatively easy to measure, also in light of the large variety of tasks available in the literature. However, while this could be seen as an advantage, it also limits the direct comparison between studies. Different procedures have been used over the years to capture delay discounting in human behavior. From “fill-in-the-blank” or “matching” questions [38] to binary choices over a fixed-set of options [13] and binary choices with an underlying staircase design optimization procedure [35,39]. All such tasks have proven to reliably detect to a certain degree the typical (or atypical, when dysregulated) decay in subjective value for monetary and other rewards [15,33,40–42]. Nevertheless, very few studies compared different methodologies or their impact on the detection of inter-individual differences [40,43,44]. Crucially, normative data to evaluate individual performances are not currently available.

The present study aims to fill this gap by proposing a version of the delay discount-ing task based on monetary reward and providing percentile equivalents from an Italian sample of 357 healthy participants, stratified by sex, age, and education. To this aim, we chose to adopt a monetary version of the delay discounting task based on a staircase pro-cedure that reliably detected delay discounting in healthy and pathological populations regardless of minor variations on the procedure or the model used to fit participants’ choices [20,35], as also confirmed by replication studies [37]. Crucially, we chose a task version that could be able to (a) provide enough data points (and, thus, indifference points) to fit the hyperbolic and exponential models and (b) that could be easily and meaningfully completed by participants in a few minutes, to overcome the crucial issue of fatigue in critical populations, such as patients or elderly participants.

  1. it is unclear how the subjects were recruited, how many were dropped, and how the authors estimated the sample size for the current study. More importantly, do the authors consider the current sample being representative of the Italian population?

We thank the reviewer for this comment. We included some additional information to the Participants section of the paper (see from line 89), where details about place and means of recruitment and inclusion/exclusion criteria are reported for each group separately.

The sample size was set to provide a reliable representation of the Italian sample. Unlike studies that aim to test a specific hypothesis or detect an effect, standardization studies do not require a power analysis, as they aim to provide a descriptive representation of the investigated behavior as accurate as possible. Thus, we aimed to recruit the largest sample accessible with our resources.

The sample can be considered representative of the Italian adult population in terms of the three main characteristics considered, namely sex (187 M - 170 F), age (min 21 - max 92 years old), and education (min 3 - max 23). A limitation to the representativeness may be constituted by the prevailing recruitment in the north of Italy (Veneto, Lombardia, and Emilia Romagna), as compared to the centre (Marche, Toscana) and south (Campania). We now acknowledge this as a limitation of our study in the Discussion section:

Line 432

Lastly, a limitation to the representativeness of the normative sample used in this study can be represented by the fact that the sample was mainly (although not exclusively) recruited in the northern regions of Italy (Veneto, Lombardia, and Emilia Romagna), although central (Marche, Toscana) and southern (Campania) regions were included (see Participants section). Crucially, the sample can be considered representative of the Italian adult population in terms of the three main characteristics considered, namely sex, age, and education age (see Table 2). Future studies may help clarify these issues by extending the evidence presented in this paper not only to other patients but also to normative samples from other nationalities.

  1. an important limitation of the study is the very small sample size of the mOFC and nonFC groups. Given this sample size, the results of the classification are likely to be unreliable.

We agree that the sample size of the patients is fairly small and we acknowledge it as a limitation of our study in the discussion section (lines 429-431).

Nevertheless, it has to be noted that this sample size is not dissimilar from that reported in other studies on this kind of population (e.g., Sellitto et al., 2010). This is mainly due to the rareness of such specific lesion. Unfortunately, there is a lack of epidemiological data on this precise population, epidemiological studies report a prevalence of 0.009% for frontotemporal degeneration (see Wada-Isoe et al. 2012, DOI: 10.1159/000342972; Pirau and Lui 2018, PMID: 30422576), 5-6% for intracranial aneurysms (Bijlenga et al., 2013,  DOI:10.1161/STROKEAHA.113.001667), 0.5% for meningiomas (Nakasu et al. 2021, DOI:10.1007/s00701-021-04919-8), 3.5% for traumatic brain injuries (Corrigan et al. 2010, DOI: 10.1097/HTR.0b013e3181ccc8b4) although estimates are based on varying sources of data, methods of calculation, and assumptions. In our sample, patients were about 3% (N=10) in respect to the normative population (N=357), thus we believe it to be fairly representative of actual prevalence in the population.

Additionally (as reported in response n. 3 of this revision letter), since we agree with the reviewer that comparing groups with different sample sizes may pose some problems, we coherently adopted a statistical model that can take into account and correct for possible imbalances amongst the variables inserted in the model, namely, sex, age, and education (see Huitema, B. 1980. The analysis of covariance and alternatives. New York: Wiley). Moreover, when using Anova, the robustness of results despite possible imbalances is ensured as long as variances are homogeneous, which was the case for our analysis (see Wilcox, R. R., Char1in, V. L., & Thompson, K. L. 1986. New monte carlo results on the robustness of the anova f, w and f statistics. Communications in Statistics-Simulation and Computation, 15-4, 933-943).

  1. for the delay discounting task, the authors should provide a graphic illustration.

We thank the reviewer for this suggestion. We added the following figure to the revised manuscript:

Figure 1. – Graphical representation of the delay discounting task. In each trial, after a 1-sec intertrial interval, participants chose between a small amount of reward delivered immediately and a larger amount of reward delivered after a delay. The preferred option remained highlighted for 1 second on the screen. See the materials and methods sections for more details.

  1. Furthermore, why did the authors not use 5 or 10 years in their task, these two delays have been frequently used in previous studies.

We are aware that other tasks in the past have included larger delays in their choice questions (e.g., Myerson et al., 2003), however, based on our experience, participants tend not to modulate their behavior when facing 1 year, 5 years, and 10 years, making the task unnecessary long (besides, current procedures tend to indeed avoid such large delays unless relevant to the question, for instance, if the reward at stake is “health”). Moreover, our staircase procedure well serves the scope of capturing hyperbolic decay in subjective value.

  1. not only significance, effect size measures should be reported, for instance, Table 3, section Discriminant validity on page 7, etc.

We agree with this observation and, thus, added effect size measures in Table 3 and through the Results section of the revised manuscript.

  1. can the authors provide some explanation or interpretation of the data reported in Table 4 and 5 in the main text?

We sincerely thank the reviewer for this comment. The previous version of the manuscript contained a description of the results from Table 4 and 5 that we now realize was limited to the Discussion section and, most importantly, was extremely poor. We, thus, expanded it in the Discussion and anticipated an explanation also in the Results section. We added the following text to the revised manuscript:

From line 290

These normative data aim to the identification of extreme performances.

For hyperbolic k scores (Table 4), the bigger the value, the higher the delay discounting [29]. Thus, k scores falling below the 10th percentile indicate extremely low performance, while k scores falling above the 90th percentile indicate extremely high performance. For instance, any female with 9 to 13 years of education scoring lower than -7.57 should be considered as presenting an extremely low delay discounting; similarly, any female with 9 to 13 years of education scoring higher than 2.73 on the hyperbolic k should be considered as presenting an extremely high delay discounting (Table 4).

The opposite interpretation should be applied to AUC.

For AUC scores (Table 5), the smaller the value, the larger the delay discounting [29]. Thus, AUC scores falling below the 10th percentile indicate extremely high performance, while k scores falling above the 90th percentile indicate extremely low performance. For instance, any male with 9 to 13 years of education scoring lower than 0.08 should be considered as presenting an extremely high delay discounting; similarly, any male with 9 to 13 years of education scoring higher than 0.79 on the hyperbolic k should be considered as presenting an extremely low delay discounting (Table 5).

From line 405

Based on these analyses, we propose that the task and normative data presented in this paper can be used to identify extremely high or low delay discounting performances. Since for hyperbolic k higher scores indicate higher delay discounting [29], extremely low or high performances are indicated by k scores falling below the 10th percentile or above the 90th percentile, respectively. For instance, any male with 9 to 13 years of education scoring lower than -6.55 should be considered as presenting an extremely low delay discounting. In opposition, since for AUC lower scores indicate higher delay discounting [29], extremely low or high performances are indicated by k scores falling above the 90th percentile or below the 10th percentile, respectively. For instance, any female with 9 to 13 years of education scoring lower than 0.02 should be considered as presenting an extremely high delay discounting (Table 5).

Round 2

Reviewer 2 Report

Thank the authors for addressing my concerns.

Author Response

Thank you as well.